# Patient-Specific Instrumentation Accuracy Evaluated with 3D Virtual Models

**DOI:** 10.3390/jcm10071439

**Published:** 2021-04-01

**Authors:** Vicente J. León-Muñoz, Andrea Parrinello, Silvio Manca, Gianluca Galloni, Mirian López-López, Francisco Martínez-Martínez, Fernando Santonja-Medina

**Affiliations:** 1Department of Orthopaedic Surgery and Traumatology, Hospital Clínico Universitario Virgen de la Arrixaca, Ctra. Madrid-Cartagena, s/n, 30120 Murcia, Spain; fcomtnez@um.es (F.M.-M.); fernando@santonjatrauma.es (F.S.-M.); 2Product Management Department, Medacta International SA, Strada Regina, 6874 Castel San Pietro, Switzerland; parrinello@medacta.ch; 3Patient Matched Technology Department, Medacta International SA, Strada Regina, 6874 Castel San Pietro, Switzerland; manca@medacta.ch (S.M.); galloni@medacta.ch (G.G.); 4Subdirección General de Tecnologías de la Información, Servicio Murciano de Salud, Avenida Central, 7, Edificio Habitamia, Espinardo, 30100 Murcia, Spain; mirindalopez@gmail.com; 5Department of Surgery, Pediatrics and Obstetrics & Gynecology, Faculty of Medicine, University of Murcia, 30100 Murcia, Spain

**Keywords:** patient matched technology, total knee arthroplasty, total knee replacement, knee, alignment, three-dimensional analysis

## Abstract

There have been remarkable advances in knee replacement surgery over the last few decades. One of the concerns continues to be the accuracy in achieving the desired alignment. Patient-specific instrumentation (PSI) was developed to increase component placement accuracy, but the available evidence is not conclusive. Our study aimed to determine a PSI system’s three-dimensional accuracy on 3D virtual models obtained by post-operative computed tomography. We compared the angular placement values of 35 total knee arthroplasties (TKAs) operated within a year obtained with the planned ones, and we analyzed the possible relationships between alignment and patient-reported outcomes. The mean (SD) discrepancies measured by two experienced engineers to the planned values observed were 1.64° (1.3°) for the hip–knee–ankle angle, 1.45° (1.06°) for the supplementary angle of the femoral lateral distal angle, 1.44° (0.97°) for the proximal medial tibial angle, 2.28° (1.78°) for tibial slope, 0.64° (1.09°) for femoral sagittal flexion, and 1.42° (1.06°) for femoral rotation. Neither variables related to post-operative alignment nor the proportion of change between pre-and post-operative alignment influenced the patient-reported outcomes. The evaluated PSI system’s three-dimensional alignment analysis showed a statistically significant difference between the angular values planned and those obtained. However, we did not find a relevant effect size, and this slight discrepancy did not impact the clinical outcome.

## 1. Introduction

Total knee arthroplasty (TKA) is a safe and effective surgical option for patients suffering from disabling knee osteoarthritis [1,2]. Over the past few decades, it has been suggested that achieving a neutral mechanical axis during surgery improved the outcome and increased the durability of TKA [3,4]. Nevertheless, other studies have reported no significant difference in survivorship among neutral mechanically aligned TKA (hip–knee–ankle (HKA) angle of 180 ± 3 degrees) compared with those that were outside that range and considered outliers [5,6,7]. These observations, together with an alternative alignment philosophy [8,9,10] that advocates the implant alignment on the knee kinematic axes that dictates the motion of the tibia and patella around the distal femoral epiphysis, justify the current debate and the lack of consensus about the optimal alignment for TKA [11,12,13].

Regardless of the optimal post-operative axes to achieve, there is no consensus on accomplishing this. The limited precision and variability in the alignment of components of the conventional instrumentation (CI) guides [14,15,16] have provided an impetus for new technology to improve surgical accuracy. First, computer-assisted surgery (CAS) was introduced. Later, patient-specific instrumentation (PSI) was developed to streamline the surgical process and increase accuracy. The latest technological innovation is robotic-assisted TKA. It has been widely published that CAS significantly improves mechanical alignment and prosthesis positioning and reduces the percentage of alignment outliers, compared to CI [17,18,19]. There are more discrepancies in the literature on whether PSI improves the accuracy of CI significantly [20]. These disagreements can even be seen in systematic reviews and meta-analyses [21,22,23,24,25,26,27,28,29,30,31,32,33,34,35,36]. However, the published meta-analyses also have limitations, as they compare the accuracy results of PSI systems that differ from each other. The different PSI systems vary depending on the image acquisition system (magnetic resonance imaging (MRI) or computed tomography (CT)) [26,37], the software used for the planning phase, and whether a pin positioner or a cutting block is designed. A given manufacturer’s PSI system’s technical characteristics cannot be strictly extrapolated to others. In contrast, the coronal alignment we achieved in the past (measured on weight-bearing full-length anteroposterior radiographs of the lower limb) with the PSI system we used [20] is comparable to that of other authors, who use the same PSI system [38,39,40,41].

Another interesting aspect to consider is that most articles emphasize accuracy in the coronal plane, but few [42,43,44,45,46,47,48] analyze the placement of the components in the sagittal and axial planes. Boonen et al. [47] and De Vloo et al. [48] assessed the accuracy of component placement in TKA using virtual 3D bone models. The authors planned the surgery with MRI, so they required a pre-operative CT and employed an iterative closest point algorithm to evaluate the post-operative CT outcome [48].

Our study aimed to determine the three-dimensional accuracy of a CT-scan-based PSI system on 3D virtual models obtained by post-operative CT. We hypothesize that the deviation of the three planes’ post-operative angular values is neither significant nor relevant.

## 2. Experimental Section

Our research was a single-center cohort study prospectively conducted and approved by the Institutional Review Board and the Ethical Committee (CEIm Hospital General Universitario José María Morales Meseguer; CPVLM 042019v2CIEST:22/19). Patients were briefed on the objectives of the study and subsequently signed an approved IRB consent document. Our study followed the World Medical Association Declaration of Helsinki’s ethical standards, as revised in 2013.

We invited all patients operated on in 2018 (January to December) by the first author of this article (senior surgeon) using MyKnee™ patient-specific TKA instrumentation (Medacta International SA, Castel San Pietro, Switzerland) to participate in the study. Thirty-one patients agreed to participate in the research (35 TKA, as 4 cases were simultaneous bilateral TKA) out of a total of 38 patients (43 TKA, with five simultaneous bilateral TKA). All surgeries were performed with the implant options of cemented fixed-bearing knee prosthesis Global Medacta Knee (GMK) Sphere (medially stabilized) (32 cases, 91.4%) and GMK Primary (ultra-congruent implant) (3 cases, 8.6%) (Medacta International SA, Castel San Pietro, Switzerland). In all cases, mechanical alignment criteria were used. Before this study, the operating surgeon had experience with over 300 TKA performed using MyKnee™ patient-specific TKA instrumentation.

All patients had undergone a pre-operative CT scan to design the operation on 3-dimensional (3D) virtual models and the PSI cutting jigs. Twelve months after surgery, we performed a new CT-scan study on the TKA and patients’ clinical evaluation. Both CT-scan studies (pre- and post-operative) were performed under the same non-weight-bearing conditions described in other studies of our group [49,50]; we employed a Somatom scope scanner (Siemens Healthcare GmbH, Erlangen, Germany) with 16 slices per rotation. The study consisted of three separate short spiral axial scans: hip, knee, and ankle. Each acquisition was centered and zoomed accurately to ensure that the field of view (FoV) maximized the region of interest. Scans were acquired in slices of a minimum of 512 × 512 pixels. A single slice thickness was 2 mm for the hip and ankle and 0.6 mm for the knee, with a maximum FoV of 200 mm. We adjusted the voltage peak to 130 kV and the X-ray tube current to 60 mA. The average effective dose of radiation per CT-scan (hip, knee and ankle) was 0.4 mSv. The MyPlanner^®^ software (Medacta International, Castel San Pietro, Switzerland) was used to create the pre- and post-operative 3D virtual models of the femur, tibia, and primary implant in situ using the CT-scans. The workflow designed by the Medacta MySolution department using Materialize’s Interactive Medical Image Control System (Mimics^®^, Materialize, Leuven, Belgium) and the tools provided by the software allow for a reduction in the dispersion of metal hardware to perform measurements with adequate accuracy. Two engineers from the Medacta MySolution department (“Engineer 1” and “Engineer 2”) independently performed the measurements on the 3D virtual models, up to 0.5 degrees of precision [24].

We used the same definition of angles and alignment criteria as in other studies [49,50]. For each case, the hip–knee–ankle (HKA) angle (angle between the femoral and the tibial mechanical axes on the medial side), the supplementary angle of the femoral lateral distal angle (sFLDA) (angle on the medial side between the mechanical axis of the femur and the femoral articular axis), the proximal medial tibial angle (PMTA) (angle on the medial side between the mechanical axis of the tibia and the tibial articular axis), the tibial slope, and the external femoral rotation measured by the condylar twist angle (CTA) (angle between the posterior condylar line and the clinical or anatomical transepicondylar axis) were determined. In post-operative measurements, each implant’s metal surface was used to reference the joint axis. Moreover, the engineers determined the slope of the tibial component (angle between the tibial slope axis (tangent to the metal surface of the tibial tray on the sagittal plane) and the tibial mechanical axis measured on the sagittal plane), the femoral flexion angle (FFA) (angle between the posterior cut plane and the femoral mechanical axis measured on the sagittal plane), the femoral rotation with the posterior condylar angle (PCA) (angle between the posterior condylar line and the surgical transepicondylar axis), the femoral version (FV) (angle between the femoral neck axis and the posterior condylar line), and the tibial torsion (TT) (angle between the line connecting the posterior cortices of the proximal tibial condyles and the line connecting the most prominent points of the medial and lateral malleolus). Figure 1 shows the measurement of angular values in the axial plane. Both engineers also assessed the three-dimensional adequacy or inadequacy of the size of the prosthetic components employed.

Twelve months after surgery, an independent observer (senior orthopedic surgeon) evaluated patients clinically, and they filled out the Forgotten Joint Score for the Knee (FJS-12) questionnaire [51], a measurement of patient-reported outcomes quantifying the patient’s ability to forget the artificial joint in everyday life. In our study, the FJ-12 survey was handed out and collected after completion in the Radiology Department on the day the X-rays and CT scan were performed, avoiding any bias due to the surgeon’s non-presence. HKA, sFLDA, and PMTA were measured on weight-bearing full-length anteroposterior radiographs of the lower limb (LLRs) by two experienced evaluators (“Evaluator 1” and “Evaluator 2”; two orthopedic surgeons different from the clinical evaluator) using the software application MicroDicom^©^ DICOM viewer for Windows and the angle measurement tools.

Statistical analysis was performed using the Statistical Package for the Social Sciences (SPSS), version 25 for Windows (SPSS, Inc., Chicago, IL, USA). Patient demographics were summarized using descriptive statistics. The Shapiro–Wilk test assessed the distribution of the data. The independent samples *t*-test and the analysis of variance were used for normally distributed variables. The one-sample *t*-test (employing for test value the planned angular value) was also applied. For nonparametric variables, the Mann–Whitney U test was utilized. Stepwise multiple regression analysis for assessing the variables that influence the FJS-12 outcomes was performed.

For the inter-observer concordance analysis, the intra-class correlation coefficient of absolute concordance was calculated using a two-factor random-effects model (ICC (2,1) [52]). We assessed intra- and inter-observer reliability according to the criteria by Landis and Koch (<0 indicate no agreement, 0.00 to 0.20 indicate slight agreement, 0.21 to 0.40 indicate fair agreement, 0.41 to 0.60 indicate moderate agreement, 0.61 to 0.80 indicate substantial agreement, and 0.81 to 1.0 indicate almost perfect or perfect agreement) [53]. Cohen’s kappa coefficient (κ) was used to measure inter-rater reliability for categorical variables. Statistical significance was reported at a *p*-value of <0.05 (two-sided). Cohen’s effect size d was calculated for all results.

## 3. Results

Thirty-five TKAs in thirty-one patients (17 female and 14 males) were analyzed. Twenty-three cases (65.7%) were on the right side, and 12 (34.3%) on the left side. Descriptive data of the series are summarized in Table 1.

### 3.1. Results on the Coronal Plane

The mean (SD (standard deviation)) pre-operative mechanical alignment (HKA angle) was 176.57° (4.44°) with a maximum varus of 10.5° and a maximum valgus of eight degrees. The mean (SD) (range) pre-operative sFLDA was 91.9° (2.21°) (86°–97°), and the mean (SD) (range) pre-operative PMTA was 87.19° (2.66°) (82°–92.5°). Accuracy results on the coronal plane are shown in Table 2.

The interobserver reliability for the engineers’ coronal measurements on the virtual TKA models was perfect: ICC (2,1) = 0.977 (CI95%: 0.954 to 0.988) for sFLDA, 0.974 (CI95%: 0.931 to 0.986) for PMTA, and 0.987 (CI95%: 0.971 to 0.993) for HKA. Comparison of their measurements using the t-test for independent variables did not yield significant differences (*p* = 0.844 for sFLDA, *p* = 0.546 for PMTA, and *p* = 0.759 for HKA). The discrepancy between the engineers’ measurements did not exceed one degree in any case, with an average of 0.34° (SD 0.32°). We observed a statistically significant difference (*p* = 0.002) when comparing the sFLDA values measured by both engineers on CT-scans with those measured by both evaluators on radiographs. There was no statistically significant difference in the comparison of either HKA values or PMTA values.

The percentage of post-operative HKA in the range 180° ± 3° was 82.86% (29/35) for Engineer 1 and 85.71% (30/35) for Engineer 2. The outliers for Engineer 1 were one case with 1° more valgus (HKA = 184°) and five cases in varus: three cases 0.5° (HKA = 176.5°), one case with 1° (HKA = 176°), and one case 2° (HKA = 175°). The outliers for Engineer 2 were one case with 0.5° more valgus (HKA = 183.5°) and four cases in varus: 0.5° (HKA = 176.5°), 1° (HKA = 176°), 1.5° (HKA = 175.5°), and 2° (HKA = 175°). The percentage of post-operative sFLDA in the range 90° ± 2° was 85.71% (30/35) for both engineers. The percentage of post-operative PMTA in the range 90° ± 2° was 80% (28/35) for Engineer 1 and 82.86% (29/35) for Engineer 2. The planned ± 3° criterion of other papers [47,54] improved the percentages to 91.43% for the sFLDA (both Engineers) and 94.29% (Engineer 1) and 97.14% (Engineer 2) for the PMTA. As shown in Table 3, the mean discrepancies between the planned and the obtained values were 1.64° ± 1.3° for HKA and less than 1.5° for sFLDA and PTMA.

The percentage of post-operative HKA in the range 180° ± 3° was 82.86% (29/35) for X-ray Evaluator 1 and 80% (28/35) for X-ray Evaluator 2. The outliers for X-ray Evaluator 1 were one case with 2° more valgus (HKA = 185°) and five cases in varus: one case with 0.5° (HKA = 176.5°), two cases with 1° (HKA = 176°), and one case 3° (HKA = 174°). The outliers for X-ray Evaluator 2 were one case with 0.5° more valgus (HKA = 183.5°), one case with 2° more valgus (HKA = 185°), and five cases in varus: one case with 0.5° more varus (HKA = 176.5°), two cases with 1° (HKA = 176°), one case with 2.5° (HKA = 174.5°), and one case with 3° (HKA = 174°).

### 3.2. Results on the Sagittal Plane

The mean (SD) pre-operative tibial slope was 81.04° (3.92°) with a range of 69.5° to 87°. The planned tibial slope was 87° (three degrees of a posterior tibial slope) for 5 cases (14.3%) and 88° for the remaining 30 (85.7%), with a mean (SD) of 87.86° (0.35°). The mean (SD) (range) post-operative tibial slope was 89.3° (2.64°) (85°–95°) according to the measurements of Engineer 1, and 89.26° (2.57°) (85.5°–95°) according to the assessment of Engineer 2 (values below 90° indicate a posterior tibial slope, and values above 90° indicate an anterior tibial slope). We considered as outliers those values with a difference of more than ±2° from the planned value. The percentage of post-operative slope in the range ±2° regarding the planned value was 62.86% (22/35) for both engineers. For both evaluators, two outliers were for slope increase (mean value out of range 0.63° (SD 0.25)) and 11 for posterior slope decrease (mean value out of range 2.31° (SD 1.69)). The planned ±3° criterion of other papers [47,54] improved the percentage of post-operative slope in the range to 82.86% (29/35). The interobserver reliability for the engineers’ slope measurements was perfect: ICC (2,1) = 0.994 (CI95%: 0.988 to 0.997).

The mean (SD) (range) planned femoral sagittal flexion angle (FFA) was 0.79° (0.98°) (0–3°). The mean (SD) (range) post-operative FFA was 1.47° (1.34°) (0–4.5°) according to the measurements of Engineer 1, and 1.37° (1.2°) (0–4.5°) according to the assessment of Engineer 2. We considered as outliers those values with a difference of more than ±2° from the planned value. The percentage of post-operative FFA in the range ±2° regarding the planned value was 91.43% (32/35) for both engineers. The planned ±3° criterion of other papers [47,54] improved the percentage of post-operative FFA in the range to 94.29% (33/35). The discrepancy between planned and measured values was caused by an increase in the FFA in all cases. The interobserver reliability for the engineers’ FFA measurements was perfect: ICC (2,1) = 0.975 (CI95%: 0.949 to 0.987).

### 3.3. Results on the Axial Plane

The interobserver reliability for the engineers’ axial measurements on the virtual TKA 3D models was perfect: ICC (2,1) = 0.971 (CI95%: 0.926 to 0.984) for CTA, 0.976 (CI95%: 0.906 to 0.987) for PCA, 0.991 (CI95%: 0.976 to 0.995) for FV, and 0.965 (CI95%: 0.902 to 0.980) for TT.

The mean (SD) pre-operative external femoral rotation measured by the CTA was 6.64° (1.77°) with a range of 3° to 9.5°. The mean (SD) planned femoral rotation (CTA) was 5.3° (2.09°) with a range of 0.5° of internal rotation (IR) to 8.5° of external rotation (ER). The mean (SD) (range) post-operative rotation (CTA) of the femoral component was 4.49° (2.43°) (1.5° IR to 8.5° ER) according to the measurements of Engineer 1, and 4.81° (2.14°) (0.5° IR to 8.5° ER) according to the assessment of Engineer 2. The planned rotation mean deviation (SD) increased by 1.56° (0.98°) and decreased by 1.64° (0.94°) for Engineer 1 and increased by 1.61° (1.41°) and decreased by 1.58° (0.92°) for Engineer 2. If we apply the criterion of planned ±3° [47], for Engineer 1, 91.43% (32/35) and Engineer 2, 94.29% (33/35) of the cases were in range. The mean (SD) (range) post-operative rotation (PCA) of the femoral component was 0.29° (2.44°) (4.5° IR to 5° ER) according to the measurements of Engineer 1, and 0.73° (2.36°) (4° IR to 5.5° ER) according to the assessment of Engineer 2. The values of the different rotational assessments are shown in Table 4.

### 3.4. Effect Size

Except for femoral rotation measured with CTA (*p* = 0.138 for Engineer 1 and 0.341 for Engineer 2), we obtained a statistically significant difference between the sFLDA, PMTA, HKA, and tibial slope planned values and those obtained post-operatively measured by both engineers. We quantified the difference’s size, as shown in Table 5.

### 3.5. Implant Sizing

In all cases, the planned femoral size was implanted. In two cases (5.71%), it was decided intraoperatively to decrease the planned tibial implant size by one size. Both engineers independently evaluated the optimal size of the implanted TKA components. The agreement was absolute (κ = 1, *p* < 0.001). For the femoral component, both considered that in 91.4% of cases, the anteroposterior size was adequate (in three cases, the anterior shield was not totally in contact), and in all cases, the mediolateral size was adequate. Regarding the tibial component, both engineers stated an adequate anteroposterior size in 82.9% of the cases (in 6 cases, they felt that the keel was too close to the cortical bone) and an adequate mediolateral size in all the cases.

### 3.6. Clinical Assessment and Patient-Reported Outcome

Thirty-four of the 35 cases were clinically assessed one year after TKA by an independent evaluator. A total of 47.1% of the cases showed no pain. Mild/occasional pain was present in 47.1%, occasional moderate pain in 5.9% and continual or severe pain in none of the patients. A total of 91.2% did not require analgesia or NSAIDs; 97.1% did not require walking aids. One patient had mediolateral instability of 6° to 9° and another (with a previous history of medial collateral ligament rupture) of 10° to 14°. All patients had adequate anteroposterior stability. No cases had extension lag. Two cases had flexion contracture less than 10°, and the mean range of flexion (SD) was 113.24° (11.73°) with a range between 80° and 130°. Only five cases had a flexion balance of less than 110°.

The mean (SD) score of the FJS-12 questionnaire (zero points = worst score and 100 points = the best score) was 57.41 points (28.78) with a range between zero and 97.92 points, a median of 64.58 points, and a percentage by quartiles of 17.1% Q1, 14.3% Q2, 45.7% Q3, and 22.9% Q4.

Multiple stepwise regression analysis showed that the variables related to FJS-12 outcomes were pain (*p* < 0.001) and flexion contracture (*p* = 0.032). Neither variables related to post-operative alignment nor the proportion of change between pre-and post-operative alignment influenced the FJS-12 results.

## 4. Discussion

Previous studies on the accuracy of component positioning with PSI showed contradictory results [21,22,23,24,25,26,27,28,29,30,31,32,33,34,35,36]. The majority of the studies use plain radiographs measurements, which may lead to inaccuracies in measurements. We are in complete agreement with Delport and Vander Sloten [56] that only by using correct techniques can the link between pre-operative planning and post-operative results be reported. Our study aimed to determine the three-dimensional accuracy of a CT-scan-based PSI system on 3D virtual models obtained by post-operative CT.

Hirschmann et al. [57] investigated the intra- and inter-observer reliability of different methods (radiographs, 2D-CT, and 3D-CT) of assessing the position of the components after TKR, and they observed that the measurements of coronal, sagittal, and rotational placement of the components using 3D-CT were highly reliable concerning inter- and intra-observer variability. Holme et al. [58] also published the superiority of 3D model measurements over plain radiographs in a unicompartmental knee replacement analysis. Based on their systematic review, De Valk et al. [59] stated that determination of component rotation after total knee arthroplasty should be performed by 3D reconstructed CT.

Jonkergouw et al. [60] described a CT-based 3D measurement method for evaluating implant positioning accuracy comparing post-operative position to the planned position using 3D virtual models of synthetic bones made from solid foam with cortical shell prepared with tantalum markers. The authors demonstrated measurement errors of less than 1.0° or 0.5 mm, and concluded that the 3D measurement method on virtual models is accurate in assessing TKA implant orientation and position in the same coordinate system as pre-operatively defined, and is independent of the planning system or the surgical implant placement technology.

To the best of our knowledge, only Boonen et al. [47] and De Vloo et al. [48] assessed the three-dimensional PSI accuracy of component placement in TKA from real patients using virtual 3D models. However, in both studies, the authors planned pre-operatively on a virtual MRI-based model, a pre-operative CT study was performed, and anatomical landmarks were extrapolated. De Vloo et al. [48] employ an iterative closest point algorithm to evaluate the post-operative CT outcome. Our analysis consisted of a direct determination on 3D models generated from post-operative CT images under identical conditions to the models generated for planning and comparing the achieved implant placement against the planned one in optimal techniques conditions.

The most important finding of the present study was satisfactory accuracy of the PSI system under evaluation in achieving the planned alignment, with a mean loss of accuracy of 1.64° ± 1.3° for the 180° planned HKA angle. Coronal plane positioning accuracy was similar for the femoral and tibial components. We observed the worst accuracy results concerning the planned values in the tibial slope. This loss of accuracy in reproducing the planned tibial slope also appeared in other articles [46,48]. In contrast, the femoral component’s sagittal alignment is very accurate, with a mean assessed discrepancy of 0.64° ± 1.09°. The results for external femoral rotation were adequate, and the implantation of the TKA did not change the torsional morphotype of the patients analyzed.

Comparing our results (both engineers’ assessments) with those published by other authors (planned value ±3 degrees) [47,54] shows an adequate alignment rate within range (84.28% for HKA, 91.43% for sFLDA, 95.71% for PMTA, 82.86% for tibial slope, 92.86% for sagittal femoral flexion, and 92.86% for the external femoral rotation measured with CTA).

Woolson et al. [46] (CT analyses of TKA post-operative alignment) state that there were no significant improvements in knee component alignment in patients treated with PSI than those treated with standard instruments. All other authors who carried out CT studies measurements [61,62,63,64,65,66,67] reported higher alignment accuracy when using PSI. A recent network meta-analysis published by Lei et al. [54] defines outliers as deviations of more than 3° from the target value and concludes that surgical robots and computer navigation improve the accuracy of alignment compared with PSI and conventional instruments in TKA. Our study is not a randomized controlled trial, so we cannot claim alignment accuracy superiority compared to other technologies. We can state that our percentage of outliers is higher than those published for robotics and navigation, very close to those published by other authors who have used the same PSI system [38,39,40], in line with previous publications [47,48], and better than those published with other systems [45,46,68,69,70,71,72].

The mean scores obtained on the FJS-12 one year after the intervention (57.41 ± 28.78, range 0 to 97.92) can be framed in the range of those published for TKA in other articles [73,74,75]. We have not established (multiple stepwise regression analysis) any relationship between the alignment obtained in the three planes or the change of alignment to the pre-operative alignment and the FJ-12 scores. The relationship’s absence between minimum deviation from the planned values or between outliers of 180° ± 3° and clinical outcomes is in line with different publications [5,6,7,54].

Technological innovation in clinical applications cannot ignore the economic aspects, namely the cost/benefit ratio. The increased cost of PSI over conventional instrumentation can be compensated by the possibility of planning before surgery with a computer-aided design virtual 3D model. Such planning can result in optimizing decisions, the absolute customization of each TKA, and the reduction of the unexpected during surgery [76].

There are some limitations to our study. First, we only evaluated PSI from one manufacturer, and thus our outcomes may not be extrapolated to other manufacturers and are not representative of the overall technology. Our results are comparable with those previously published with different analysis types using the same PSI system [20,38,39,40,41]. Secondly, streak artefacts caused by metal implants can degrade the post-operative CT-image quality and limit three-dimensional modelling and measurements accuracy. The Medacta MySolution department has designed a nine-step workflow to manage cases with metal hardware in place using Materialize’s Interactive Medical Image Control System (Mimics^®^, Materialize, Leuven, Belgium) and the tools provided by the software to reduce the metallic hardware scattering [49]. Thirdly, we have conducted a study with a small number of cases (*n* = 35 knees). Nevertheless, on the one hand, the ICC values give some confidence that the results will be more widely applicable and, on the other hand, by the inclusion criterion of surgeries performed by a single surgeon with PSI in the one year, we avoided any selection bias. Fourthly, alignment assessment was performed with supine non-weight-bearing CT-scan-based 3D models. Previous studies [50] have shown significant discrepancies among weight-bearing LLRs and supine non-weight-bearing CT-scan-based 3D models in assessing the knee joint alignment before TKA, despite a good correlation. However, we consider that the absence of loading does not invalidate our results, as both pre- and post-operative studies were performed in the same way in non-weight-bearing conditions. Finally, we lack a control group (conventional instrumentation or computer-assisted surgery) with which to compare. Methodologically, we designed the study to compare the 3D angular values obtained with those planned. It would certainly be interesting to evaluate other instrumentation systems’ accuracies with the same protocol, but this was beyond our study’s purpose.

The last decade has seen remarkable advances in knee replacement surgery. Any new technology’s value depends on its potential to improve outcomes compared to technologies already established. There is no evidence that achieving PSI alignment targets is superior to that of other systems. However, we must not forget that some parameters are difficult to objectify through RCTs and meta-analyses. The ability to plan in 3D, the flexibility to adapt to any alignment philosophy, the information on the implant sizes to be used, and other logistical advantages make TKA surgery a more patient and surgical team “friendly process” and are additional advantages of PSI that should be considered. Recent studies show no clinical significance in postoperative outcomes between conventional instruments, PSI, navigation, and robots [54]. Therefore, the above-mentioned additional advantages, which are difficult to quantify, and the adequacy of the costs, may influence the surgeon’s decision to use PSI as an instrumentation system. When using PSI technology, it should not be forgotten that there is also a certain margin of inaccuracy due to the human factor (a surgeon places the blocks on the femur and tibia and performs the bone cuts), even more so when dealing with margins as narrow as one or two degrees.

## 5. Conclusions

The evaluated PSI system’s three-dimensional alignment analysis shows a statistically significant difference between the angular values planned and those obtained. However, we did not find a relevant effect size. Nor has the slightest discrepancy between planned and achieved had any clinical impact.

## Figures and Tables

**Figure 1 jcm-10-01439-f001:**
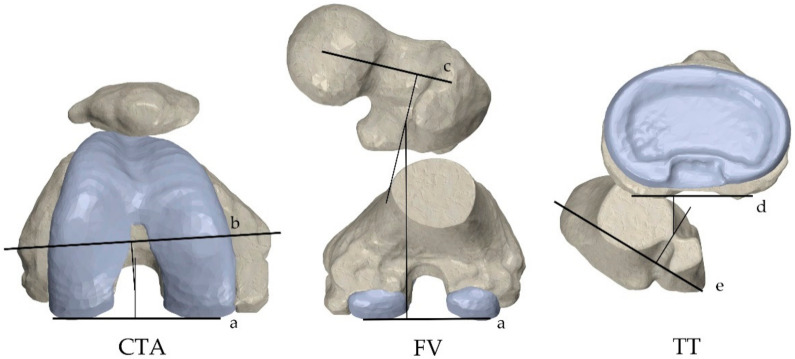
Measurement of angular values in the axial plane on the 3D virtual model of the total knee arthroplasty (TKA). CTA: the condylar twist angle is the angle between the posterior condylar line (**a**) and the clinical transepicondylar axis (**b**). FV: the femoral version is the angle between the femoral neck axis (**c**) and the posterior condylar line (**a**). TT: the tibial torsion is defined as the angle between the line connecting the posterior cortices of the proximal tibial condyles (**d**) and the line connecting the most prominent points of the medial and lateral malleolus (**e**).

**Table 1 jcm-10-01439-t001:** Series descriptive data.

	Weight (kg)	Height (cm)	BMI (kg/m^2^)	Age (Years)
Female cases	71.1 (6.9) (57–84)	159.3 (4.3) (149–165)	28.1 (3.1) (21.7–33.2)	72.3 (5.8) (59–81)
Male cases	75 (6.9) (63–84)	165.2 (6) (157–175)	27.5 (2.3) (22.6–32.1)	69.8 (7.5) (57–81)

Mean value (standard deviation) (range). BMI: body mass index.

**Table 2 jcm-10-01439-t002:** TKA-determined angular values. Results are shown as mean value (standard deviation) and (range). Engineers 1 and 2 performed the measurements on the 3D CT-scan-based virtual models. Evaluators 1 and 2 measured weight-bearing full-length anteroposterior radiographs of the lower limb. sFLDA: supplementary angle of the femoral lateral distal angle, PMTA: proximal medial tibial angle, HKA: hip–knee–ankle angle.

	sFLDA (°)	PMTA (°)	HKA (°)
Engineer 1	90.96 (1.52) (88–94.5)	88.73 (1.34) (86–92)	179.09 (1.95) (175–184)
Engineer 2	91.03 (1.5) (88.5–94.5)	88.91 (1.22) (86–92)	179.23 (1.93) (175–183.5)
Mean CT-scan	90.99 (1.5) (88–94.5)	88.82 (1.27) (86–92)	179.16 (1.93) (175–184)
Evaluator 1 ^#^	90.19 (1.43) (87–94)	88.7 (1.13) (86.5–92)	179.1 (2.34) (174–185)
Evaluator 2 ^#^	90.16 (1.67) (86–94)	88.69 (1.08) (86.5–91.5)	179.16 (2.33) (174–185)
Mean X-ray ^#^	90.17 (1.55) (86–94)	88.69 (1.1) (86.5–92)	179.13 (2.32) (174–185)

^#^ The results shown correspond to the evaluators’ first radiographic measurements. There were no statistically significant differences between their two separate measurements in time, and both evaluators showed perfect intra-observer reliability in all angular values (ICC (2,1) between 0.953 and 0.995). We also found no difference between the measurements of both X-ray evaluators (*p* = 0.939 for sFLDA, *p* = 0.957 for PMTA and *p* = 0.877 for HKA) and a perfect inter-observer reliability: ICC (2,1) = 0.974 (CI95%: 0.949 to 0.987) for sFLDA, 0.945 (CI95%: 0.849 to 0.973) for PMTA and 0.989 (CI95%: 0.979 to 0.995) for HKA.

**Table 3 jcm-10-01439-t003:** Discrepancies observed between the planned and achieved values on the 3D CT-scan-based virtual models.

	sFLDA (°)	PMTA (°)	HKA (°)
Engineer 1	1.44 (1.05) (0–4.5)	1.56 (0.98) (0–4)	1.71 (1.28) (0–5)
Engineer 2	1.46 (1.08) (0–4.5)	1.31 (0.96) (0–4)	1.57 (1.34) (0–5)
Mean values	1.45 (1.06) (0–4.5)	1.44 (0.97) (0–4)	1.64 (1.3) (0–5)

Results are shown as mean value (standard deviation) and (range). sFLDA: supplementary angle of the femoral lateral distal angle (90° planned), PMTA: proximal medial tibial angle (90° planned), HKA: hip–knee–ankle angle (180° planned).

**Table 4 jcm-10-01439-t004:** Axial measurements on the virtual TKA 3D CT-scan-based virtual models. Results are shown as mean value (standard deviation) and (range). FV: femoral version, TT: tibial torsion. Negative values for FV indicate femoral retroversion, while positive values indicate femoral anteversion. *n* = 28 (we lacked pre-operative information in seven cases since FV and TT are not standardized angular values in surgical planning).

	Pre-Operative	Engineer 1 *	Engineer 2 *
**FV (°)**	12.79 (7.17) (−2.5–31.5)	10.32 (7.49) (−2.5–26.5)	10.91 (7.16) (−1–27)
**TT (°)**	24.6 (7.76) (6.5–39)	24.09 (7.33) (10–36)	25.3 (7.54) (13.5–38.5)

* Both evaluators showed a perfect inter-observer reliability: ICC (2.1) = 0.990 (CI95%: 0.974 to 0.995) for FV and 0.963 (CI95%: 0.900 to 0.981) for TT. We obtained no significant differences between the two engineers’ values or between these values and the patients’ constitutional pre-operative values.

**Table 5 jcm-10-01439-t005:** Quantification of effect size using Cohen’s d. sFLDA: supplementary angle of the femoral lateral distal angle, PMTA: proximal medial tibial angle, HKA: hip–knee–ankle angle, FFA: femoral sagittal flexion angle, CTA: condylar twist angle.

	d-Value for Engineer 1	d-Value for Engineer 2
Planned vs. measured sFLDA	−0.89	−0.97
Planned vs. measured PMTA	1.35	1.26
Planned vs. measured HKA	0.66	0.56
Planned vs. measured FFA	−0.58	−0.53
Planned vs. measured tibial slope	−0.76	−0.76
Planned vs. measured CTA	0.36	0.32

d-value to qualify the magnitude of an effect (the difference between means) can be interpreted according to the criteria by Hopkins et al. [55]: less than 0.2, trivial; 0.2 to 0.59, small; 0.6 to 1.19, moderate; 1.20 to 2, large; 2 to 3.99, very large, and greater than 4, extremely large.

## Data Availability

No additional data available.

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
