# Peer review of "Patient-Specific Instrumentation Accuracy Evaluated with 3D Virtual Models"

_jcm, 2021, doi:10.3390/jcm10071439_

Round 1

Reviewer 1 Report

Overall this is a good study with interesting results. The discussion and conclusions can be adjusted and augmented to bring attention to the clinically relevant results and how this data compares to the existing literature (given the limitation of not having a control group). Please see specific comments below. 

Abstract: 

Well written highlighting the important aspects of the manuscript. The limitation/conclusion section could be improved by removing the limitations and focusing more on the overall conclusions to be drawn from the study and how these results should be interpreted for changes in practice. Does this study show PSI to be better than conventional instrumentation for achieving alignment goals? 

Introduction: 

Good introduction that discusses the background of why the study was completed and the previous literature. 

Line 56: disruptive should be changed to something else as this puts a negative connotation on kinematic alignment. 

Line 63: what is meant by daily? I think this can be removed

Methods: 

Methods are well written with good description of the measurements done.

Where the two reviewers who conducted the measurements the same engineers mentioned in 117-119? If this is the case it should be specifically mentioned as their measurement accuracy was described in 117-119. If this 0.5 degrees of precision is from a previous study it should be sited. 

It looks like both engineers and separate evaluators completed the measurements. This should be stated in the methods section that both of these groups made measurements and they were compared. Should also mention differences between the engineer and evaluator measurements (ct vs xray).

line 149: "besides" can be removed

Results: 

Well written results with the most interesting being the difference between planned and actual alignment. 

The results from table 3 should be mentioned in the body of the results section. 

Table 2: Should include the mean/std deviation/range for all combined measurements as discussed in the results section. 

Discussion: 

The most interesting part of this paper is the difference between planned and actual alignment. There should be a paragraph comparing the results from this study to results from similar studies using conventional instrumentation, computer navigation, and robotics. Is PSI better, worse, or similar to these other techniques. 

There should also be a discussion paragraph discussing the clinical relevance of being 1-2 degrees off from planned. This is somewhat discussed in the introduction but should be talked about here. Does 1-2 degrees matter clinically? Does >3 degrees matter clinically? How does this study add to this literature. 

Lines 353-355 should be in the methods. 

The main limitation of this study is the lack of a comparison group. Therefore as stated above there needs to be a discussion on how this data compares to the data in the literature about other technologies. What are the benefits and negatives about navigation/robotics/conventional instrumentation compared to PSI? 

Line 389: knee bones should be changed to femur and tibia.

PSI is expensive compared to conventional instrumentation. This should be discussed as well. Is it worth the extra cost to use PSI in the context of these results? 

Conclusions: 

This should focus on the differences between planned and actual measurements and how this compares to the existing literature especially conventional instrumentation. Does PSI give a better result than conventional instrumentation? The limitations are in the discussion and do not need to be in the conclusions.

Reviewer 2 Report

JCM 1157093

Title: Patient-specific instrumentation accuracy evaluated with 3D virtual models

The authors propose to evaluate the effectiveness of a new instrument developed to increase the accuracy of the placement of the components of a knee prosthesis. They report in their study that the variables related to postoperative alignment and that the proportion of the change between preoperative and postoperative alignment do not influence the results reported by patients.  The main conclusion drawn from their study is that the accuracy of the PSI system evaluated in 3D virtual models has been satisfactory, with a limitation the rotation of the femoral component.it is an original and well written paper. Just a few comments.

(NB: continuous line numbers are better for reviewers).

Point 1: Page 2, lines 46-47. The keywords may not appear in the title.

Point 2, Page2 lines 86-87. I suggest that the authors indicate that they have met the criteria defined in the Declaration of Helsinki and indicate the name of the institution that has approved the study.
